# Identification and characterization of archaeal pseudomurein biosynthesis genes through pangenomics

Valérian Lupo,[1,2] Célyne Roomans,[1] Edmée Royen,[1] Loïc Ongena,[1] Olivier Jacquemin,[1] Coralie Mullender,[1] Frédéric Kerff,[2] Denis Baurain[1]

**ABSTRACT**    The peptidoglycan (PG, or murein) is a mesh-like structure, which is made of glycan polymers connected by short peptides and surrounds the cell membrane of nearly all bacterial species. In contrast, there is no PG counterpart that would be universally found in Archaea but rather various polymers that are specific to some lineages. Methanopyrales and Methanobacteriales are two orders of Euryarchaeota that harbor pseudomurein (PM), a structural analog of the bacterial PG. Owing to the differences between PG and PM biosynthesis, some have argued that the origin of both polymers is not connected. However, recent studies have revealed that the genomes of PM-containing Archaea encode homologs of the bacterial genes involved in PG biosynthesis, even though neither their specific functions nor the relationships within the corresponding inter-domain phylogenies have been investigated so far. In this work, we devised a pangenomic bioinformatic pipeline to identify proteins for PM biosynthesis in Archaea without prior genetic knowledge. The taxonomic distribution and evolutionary relationships of the candidate proteins were studied in detail in Archaea and Bacteria through HMM sequence mining and phylogenetic inference of the Mur domain-containing family, the ATP-grasp superfamily, and the MraY-like family. Our results show that archaeal muramyl ligases are of bacterial origin but diversified through a mixture of horizontal gene transfers and gene duplications. However, in the ATP-grasp and MraY-like families, the archaeal members were not found to originate from Bacteria. Our pangenomic approach further identified five new genes potentially involved in PM synthesis and that would deserve functional characterization.

**IMPORTANCE** *Methanobrevibacter smithii* is an archaea commonly found in the human gut, but its presence alongside pathogenic bacteria during infections has led some researchers to consider it as an opportunistic pathogen. Fortunately, endoisopeptidases isolated from phages, such as PeiW and PeiP, can cleave the cell walls of *M. smithii* and other pseudomurein-containing archaea. However, additional research is required to identify effective anti-archaeal agents to combat these opportunistic microorganisms. A better understanding of the pseudomurein cell wall and its biosynthesis is necessary to achieve this goal. Our study sheds light on the origin of cell wall structures in those microorganisms, showing that the archaeal muramyl ligases responsible for its formation have bacterial origins. This discovery challenges the conventional view of the cell-wall architecture in the last archaeal common ancestor and shows that the distinction between "common origin" and "convergent evolution" can be blurred in some cases.

**KEYWORDS**    pseudomurein, peptidoglycan, cell wall, synteny, Mur ligases, ATP-grasp, MraY-like, inter-domain HGT, phylogenetic inference

**Peer Reviewer** Damien P. Devos, UPO, Seville, Andalucia, Spain

Address correspondence to Denis Baurain, denis.baurain@uliege.be.

The authors declare no conflict of interest.

See the funding table on p. 21.

The cell wall is a complex structure that surrounds most prokaryotic cells, protects them against the environment, and maintains their internal turgor pressure (1, 2). It also constitutes one of the striking phenotypic differences between Archaea and Bacteria. Indeed, while most archaeal species possess a paracrystalline protein surface layer (S-layer) (3), other species harbor a large variety of cell-wall polymers (e.g., sulfated heteropolysaccharides, glutaminylglycan, methanochondroitin) (2, 4), whereas nearly all bacterial cell walls contain a single common polymer termed peptidoglycan (PG, also known as murein) (1, 5). PG is a net-like polymer formed by long glycosidic chains of alternating N-acetylglucosamine (GlcNAc) and N-acetylmuramic acid (MurNAc) units linked by a β-(1→4) bond. MurNAc is attached to a short peptide, from three to five amino acids (AAs) long, usually composed of L-alanine (L-Ala), D-glutamic acid (D-Glu), meso-diaminopimelic acid (meso-DAP), or L-lysine (L-Lys), and two D-alanines (D-Ala). This short peptide serves as a bridge between two glycosidic chains and is built at the final stage of PG biosynthesis (1, 5). Interestingly, there exists an archaeal cell-wall polymer that shows a three-dimensional structure similar to PG, hence named pseudo-peptidoglycan or pseudomurein (PM). Compared with PG, PM contains N-acetyl-L-talo-saminuronic acid (TalNAc) units linked to GlcNAc through a β-(1→3) bond, instead of MurNAc, and only has L-amino acids attached to TalNAc (2, 6, 7). Depending on the taxon, both PG and PM can show variation in their amino acids and glucidic composition (1, 2, 5, 6). In the early 1990s, a PM biosynthesis pathway was proposed (7–9), and due to differences between PG and PM biosynthesis, it was concluded that both polymers had evolved independently (4, 10, 11). In contrast to the ubiquity of PG, PM is found only in two orders of Euryarchaeota: Methanopyrales and Methanobacteriales. In recent phylogenomic reconstructions, Methanopyrales and Methanobacteriales are both monophyletic and further form a clade with Methanococcales as an outgroup, all three orders being collectively termed class I methanogens (12, 13). Unlike Methanopyrales and Methanobacteriales, the cell wall of Methanococcales is composed of an S-layer and does not contain PM. This restricted taxonomic distribution suggests that PM has appeared in the last common ancestor of these two orders of methanogens, after their separation from the Methanococcales lineage, and thus that PM was not a feature of a more ancient archaeal ancestor. In other studies, Methanopyrales is basal to the whole clade of class I methanogens (13, 14), which would point to a loss of PM in Methanococcales. However, it has been proposed that the latter topology might be caused by a phylogenetic artifact (15, 16).

Regarding PG, it is so crucial for cell survival and growth that even bacteria once thought to lack PG, like Planctomycetes or Chlamydiae, were actually shown to synthesize a thin layer of PG, notably during septal division (17–21). Therefore, the proteins involved in PG biosynthesis have been extensively studied over the last years, in particular as potential targets for antimicrobial agents (22). Usually, many genes involved in PG biosynthesis lie in the division and cell-wall synthesis (*dcw*) gene cluster. The order of the genes within this cluster is relatively well conserved across the different bacterial lineages (23–25), even if some species lack one or more PG biosynthesis genes in their genome (26, 27). A recent reconstruction of the ancestral state of the *dcw* cluster showed that the last bacterial common ancestor (LBCA) had a complete *dcw* cluster, probably composed of 17 genes (28).

Among the proteins encoded by *dcw* cluster genes, the four muramyl ligase enzymes (MurC, MurD, MurE, and MurF), and the D-alanine–D-alanine ligase (Ddl) are critical for PG biosynthesis. The muramyl ligases add, respectively and successively, L-Ala, D-Glu, meso-DAP (or L-Lys depending on the species) and D-Ala–D-Ala to UDP-Mur-NAc, while Ddl binds two D-Ala to yield the D-Ala–D-Ala dipeptide (1, 29). Inhibiting one of those genes leads to lysis of the bacterial cell (30, 31). The muramyl ligases belong to the ATP-dependent Mur domain-containing family, which further includes four other enzymes: (i) MurT, which in *Staphylococcus* forms a complex with GatD to catalyze the amidation of D-Glu to D-glutamine (D-Gln) (32, 33), (ii) CapB, which plays a role in the formation of the poly-γ-glutamic acid, a polymer mainly found in *Bacillus*

species, notably as the main component of the capsule of *Bacillus anthracis* (34–36), (iii) cyanophycin synthetase (CphA), which catalyzes the polymerization of L-arginine (L-Arg) and L-aspartate (L-Asp) into cyanophycin, a polymer that constitutes a nitrogen reserve in Cyanobacteria (37, 38), (iv) folylpolyglutamate synthase (FPGS), which is responsible for the addition of polyglutamate to folate. The FPGS enzyme is found in the three domains of life: Archaea, Bacteria, and Eukarya, but not in methanogenic archaea (39–41). Ddl is part of the ATP-grasp superfamily, including at least 21 groups of enzymes (42), among which the large subunit (termed CarB) of carbamoyl phosphate synthetase (CPS (43). CPS is a well-studied enzyme that has been used to root the "three-domain" tree of life because CarB results from an internal gene duplication that occurred before the last universal common ancestor (LUCA) (44–46). Nowadays, the most commonly held view is a "two-domain" tree of life with monophyletic Bacteria and paraphyletic Archaea from which emerge Eukarya (47), even though other models, such as a "one-domain" tree of life in which LUCA was a bacterium (48), are still possible.

With the advances in genome sequencing, homologs of genes involved in PG biosynthesis, including muramyl ligases, have been identified in Methanopyrales and Methanobacteriales (49–52). Consequently, it was suggested that, despite the difference between the two biosynthetic pathways, the evolution of PG and PM are evolutionarily related, probably through horizontal transfers (HGTs) of PG biosynthetic genes from Bacteria to Archaea (53–55). Hence, Subedi and co-workers used similarity searches to re-investigate the pathway proposed by Leahy et al. (52). They further resolved the structures of two archaeal muramyl ligases, which they named pMurC and pMurE, after their supposed homology with bacterial MurC and MurE, respectively (54, 56). While these two recent studies increased the catalog of genes potentially involved in PM biosynthesis, the homology-based approach assumes that the archaeal pathway is similar to what is known in bacteria. Consequently, the enlarged catalog may miss key genes that are not homologous to those of the bacterial pathway while including paralogous genes that are actually not involved in PM biosynthesis (and not specific to PM-containing archaea).

In the present work, we used a pangenomic approach that did not rely on prior genetic knowledge to identify candidate genes for PM biosynthesis. Then, we characterized their functional domains using various prediction software and assessed the taxonomic distribution of their homologs in both bacterial and archaeal domains. We also investigated the evolutionary origins of shared PM and PG biosynthesis genes, with the aim to distinguish between two main hypotheses for the emergence of PM in class I methanogens: vertical inheritance from LUCA, followed by losses in most archaeal lineages or convergent evolution through HGT from Bacteria. To this end, we performed phylogenetic analyses of the Mur domain-containing family, ATP-grasp superfamily, and MraY-like family, using multiple variations of the taxon sampling and different AA substitution models. Our results support a bacterial origin of the four main archaeal muramyl ligases, which probably traces back to two HGT events in an ancestor of Methanopyrales and Methanobacteriales, one of these transfers being followed by two rounds of gene duplication. Yet, we arrive at different conclusions than Subedi and co-workers regarding the specific homology (and naming) of pMurC and pMurE. Finally, we identified a third gene transfer responsible for the occurrence of the complex MurT and GatD into most PM-containing archaea.

## RESULTS

### Collection of candidate proteins for pseudomurein biosynthesis

For the identification of candidate genes for pseudomurein (PM) biosynthesis following a pangenomic approach independent of known genes, we used the whole proteomes of 10 archaeal organisms, corresponding to five PM-containing archaea (i.e., four Methanobacteriales and one Methanopyrales) and five non-PM Euryarchaeota (i.e., one Methanococcales, two representatives from different orders of Methanomicrobia, one Archaeoglobales, and one Thermoplasmatales). The protein sequences of the 10 archaeal

assemblies were first clustered into 6,321 orthologous groups (OGs; clusters named from OG0000001 to OG0006321). A taxonomic filter allowed us to select 82 OGs specific to the PM-containing archaea, among which 26 OGs contained sequences of all five PM-containing archaea, whereas 56 OGs contained sequences of the only Methanopyrales and three Methanobacteriales (retained to maximize the sensitivity of our search). No OG was specific to the four Methanobacteriales. The paralog-targeting approach (see Materials and Methods) allowed us to identify 20 additional OGs. In parallel, eight OGs were selected using three pseudomurein-related HMM profiles downloaded from the NCBI Conserved Domain Database (CDD, see Materials and Methods). Overall, 110 OGs were thus identified as candidates for PM biosynthesis (Fig. S1).

## Genetic environment of candidate proteins

Synteny analysis revealed that 22 out of 110 OGs are encoded by genes clustered in five regions of the genomes of PM-containing archaea, which we termed clusters A to E (Fig. S2). *In silico* functional analysis indicates (Table S1; sheets 1 to 3) that proteins of clusters A and B may be involved in PM biosynthesis while proteins of clusters C, D, and E are probably not. Cluster C is a bidirectional cluster, for which HMM-based annotations suggest that the corresponding proteins belong to at least two pathways unrelated to PM: OG0001177 and OG0001178 are associated with pilus assembly proteins and/or surface proteins, while OG0001176 and OG0000094 can be associated with cell shape or gene regulation (the latter is not identified in our pipeline but its gene is always located downstream of the OG0001176 gene). Cluster D is related to nucleic acid metabolism or cellular signal transduction (57), whereas cluster E code for the four proteins that compose the methyl-coenzyme M reductase, which is implied in methane formation (58). Clusters A and B correspond to the two clusters identified by Subedi et al. (54), based on similarity searches using bacterial PG biosynthetic proteins as queries. Cluster A is composed of five genes: (i) OG0001014, which was experimentally characterized as the smallest CPS (59), (ii) OG0001163, a type 4 glycosyltransferase homolog to MraY, (iii) OG0001473, a Mur domain-containing protein, (iv) OG0001162, and (v) OG0001472, two hypothetical proteins devoid of known domains. Regarding cluster B, it is composed of three genes: (i) OG0001150, a Mur domain-containing protein, (ii) OG0001147, another uncharacterized hypothetical protein, and (iii) OG0001146, a MobA-like NTP transferase domain-containing protein. In addition, two genes of Mur domain-containing proteins (i.e., OG0001148 and OG0001149) can be located either in cluster A or B, and even outside any cluster, depending on the PM-containing species considered. Furthermore, another PM-specific gene (OG0000796, coding for a hypothetical protein) is located just downstream of the OG0001472 gene in the genome of *Methanopyrus* sp. KOL6, while a second one (OG0000169, coding for a Zn peptidase) is only three genes away from the OG0001146 gene in *Methanothermobacter thermautotrophicus* str. Delta. Based on the genetic environment of clusters A and B, we attempted to identify a conserved regulon for PM biosynthesis by phylogenetic footprinting (60, 61). However, unlike in Bacteria (61), such analyses were unsuccessful on our archaeal data set (see Supplemental data).

Taking into account OG0000094, identified by its conserved localization within cluster C, our pipeline recovered 23 syntenic genes (out of 111 OGs), of which half are likely to be involved in PM biosynthesis (Table 1). For clarity, in the following, the four Mur domain-containing proteins OG0001148, OG0001149, OG0001150, and OG0001473 will be arbitrary called Murα, Murß, Murγ, and Murδ, respectively, without considering any specific homology with bacterial MurCDEF. Among the proteins encoded in clusters A and B, only OG0000169 and OG0001162 feature a Sec signal peptide (SP), which suggests that they are exported outside the cell. All other proteins lack such a SP, and thus are either cytoplasmic or transmembrane (TM) proteins. TM segment prediction was used to distinguish between these two cases, which revealed that only OG0000796, OG0001163,

and OG0001472 are TM proteins. Overall, these targeting analyses are useful to gain insight into the potential position of each protein in the PM biosynthetic pathway.

## Taxonomic distribution of candidate proteins and their homologs

To ensure the completeness of the selected OGs, we looked for corresponding pseudo-genes or mispredicted proteins in the genomes of the five PM-containing archaea (see Materials and Methods). As an illustration, in *Methanothermobacter thermautotrophicus* str. Delta, OG0001472 and Murδ had been annotated as pseudogenes. However, in PM-containing archaea, the synteny of these genes from cluster A is highly conserved. Thus, our "loose" pangenomic approach rightfully led us to rescue OGs that would have been missed by a stricter approach. After trying to complete all the OGs, we retained only those containing protein sequences from all five PM-containing archaea, decreasing their number from 111 to 49. Interestingly, no OG from the five syntenic regions had to be discarded. Homology searches in three local databases showed that 15 OGs are widespread (though not universal) among Bacteria and Archaea, nine OGs have homologs only in bacteria and PM-containing archaea, while 25 OGs are exclusive to archaea, among which 15 to PM-containing archaea (Fig. 1; for details see Table S2). In clusters A and B, which likely encode proteins involved in PM biosynthesis, five OGs are exclusive to Methanopyrales and Methanobacteriales, whereas seven OGs share homology with bacterial proteins. We also noticed that our HMM profiles of the four muramyl ligases (i.e., Murα, Murß, Murγ, and Murδ) recovered a common set of sequences, indicating that Murαßγδ are specifically related. According to this taxonomic distribution, we further investigated the origin of OG0001014 (currently annotated as CPS), the MraY-like and the four muramyl ligases Murαßγδ. The MobA-like NTP transfer-ase (OG0001146) was not considered for phylogenetic analyses because, compared with the aforementioned proteins, no homologous protein was identified in the representative bacterial database (nor for OG0001215 and OG0000138). However, some bacterial homologs were recovered when we determined the taxonomic distribution of the 49 OGs using the (much larger) prokaryotic database.

## Phylogenetic analyses

### ATP-grasp superfamily

OG0001014 from the cluster A of PM-containing archaea and the Ddl from the *dcw* cluster of bacteria are member proteins of the ATP-grasp superfamily. Due to the large number of protein functions and architectures within the ATP-grasp superfamily (42), we focused our phylogenetic analyses on the ATP-grasp domain. Furthermore, we wanted to investigate whether OG0001014 is actually a CPS, as proposed by Popa et al. (59), or closely related to Ddl (through HGT for instance). Thus, we excluded eukaryotic ATP-grasp proteins from our analyses. In the local databases, we identified 8,013 unique protein sequences containing at least one ATP-grasp domain, which are distributed across 1,387 prokaryotic organisms. ATP-grasp domains were spliced out of full-length proteins, yielding a total of 12,074 domain sequences, then sequence deduplication led to 2,344 sequences from which 149 highly divergent sequences were removed. Annotation showed that 1,788 domain sequences correspond to 17 members of the ATP-grasp superfamily, while 407 sequences have no similarity with reference ATP-grasp sequences (see Materials and Methods). We also observed that PyC, PccA, and AccC reference sequences annotate proteins belonging to the same monophyletic group. These three enzymes use hydrogenocarbonate as a substrate (62–64), which could explain the phylogenetic proximity of their ATP-grasp domain sequences. Accordingly, we decided to indistinctly tag the whole group with the three annotations. Similarly, PurK and PurT reference sequences mostly annotated the same set of proteins, albeit PurK uses hydrogenocarbonate as its substrate and PurT formate (65, 66).

Due to an internal gene duplication that occurred before LUCA (44–46), the seven phylogenetic trees (see Materials and Methods) were rooted on CarB, the monophyly

**TABLE 1** Overview of the proteins identified in our pangenomic search for genes involved in PM biosynthesis*[a]*

| Cluster | Orthologous groups | InterProScan prediction | Signal peptide | # TM |
|---|---|---|---|---|
| A | OG0001014 | CPS large subunit, ATP-grasp | Other | 0 |
| | OG0001163 | MraY-like | Other | >1 |
| | OG0001473 | Muramyl ligase (= Murδ) | Other | 0 |
| | OG0001162 | / | Sec | 0 |
| | OG0001472 | / | Other | 1 |
| | OG0000796 | / | Other | >1 |
| B | OG0001150 | Muramyl ligase (= Murγ) | Other | 0 |
| | OG0001147 | / | Other | 0 |
| | OG0001146 | MobA-like NTP transferase domain | Other | 0 |
| | OG0000169 | / | Sec | 0 |
| A-B | OG0001148 | Muramyl ligase (= Murα) | Other | 0 |
| | OG0001149 | Muramyl ligase (= Murß) | Other | 0 |
| C | OG0001210 | Aminotransferases class-I pyridoxal-phosphate attachment site | Other | 0 |
| | OG0000094 | MreB/DnaK-like | Other | 0 |
| | OG0001176 | Coiled coil protein | Other | 0 |
| | OG0001177 | Flp pilus assembly protein RcpC/CpaB | Other | 1 |
| | OG0001178 | Sortase E | Other | >1 |
| D | OG0001213 | Zc3h12a-like ribonuclease NYN domain | Other | 0 |
| | OG0001214 | Nucleotide cyclase | Other | >1 |
| E | OG0000266 | Methyl-coenzyme M reductase, beta subunit | Other | 0 |
| | OG0000231 | Methyl-coenzyme M reductase operon protein D | Other | 0 |
| | OG0000230 | Methyl-coenzyme M reductase, gamma subunit | Other | 0 |
| | OG0000229 | Methyl-coenzyme M reductase, alpha subunit | Other | 0 |

*[a]*Orthologous groups (OGs) composing the identified gene clusters, named clusters A to E are listed. For each OG, there is the functional prediction of InterProScan (if any), the predicted signal peptide type (SP) and the number of predicted transmembrane (TM) segments (0 = cytoplasmic, 1 = monotopic, >1 = polytopic).

of which is supported by high statistical values. Despite a low topology conservation between the different evolutionary models and number of tree search iterations, some recurring patterns can be observed (Fig. 2; Fig. S3 to S8). RimK ATP-grasp domain sequences are always paraphyletic, due to the inclusion of GshB, GshAB, and CphA, the latter two clustering into a smaller clan. The monophyly of Acetate–CoA ligases AcD (67) is maximally supported, and a long branch is present at the base of the group. Except for the C40 model (Fig. S5 and S6), AcD forms a clan with the Succinate–CoA ligase SucC (68). The position of the other members of the ATP-grasp superfamily is much more elusive. For example, Pur2 (69) emerges from an unannotated region of the LG4X tree (Fig. 2), whereas it forms a clan with either AcD and SucC in the four C20 and C60 trees (Fig. S3 and 4; S7 and 8) or only with SucC in the two C40 trees (Fig. S5 and S6). Similarly, albeit Ddl and OG0001014 branch together in one C20 tree with a branch support of 63 (Fig. S3), their respective positions within the ATP-grasp superfamily are unstable (Fig. 2; Fig. S3 to S8). In conclusion, OG0001014 is clearly not a CPS since it is never close to CarB in our trees. Moreover, there is no strong phylogenetic evidence for a specific relationship between OG0001014 and Ddl, in contrast to what we had hypothesized.

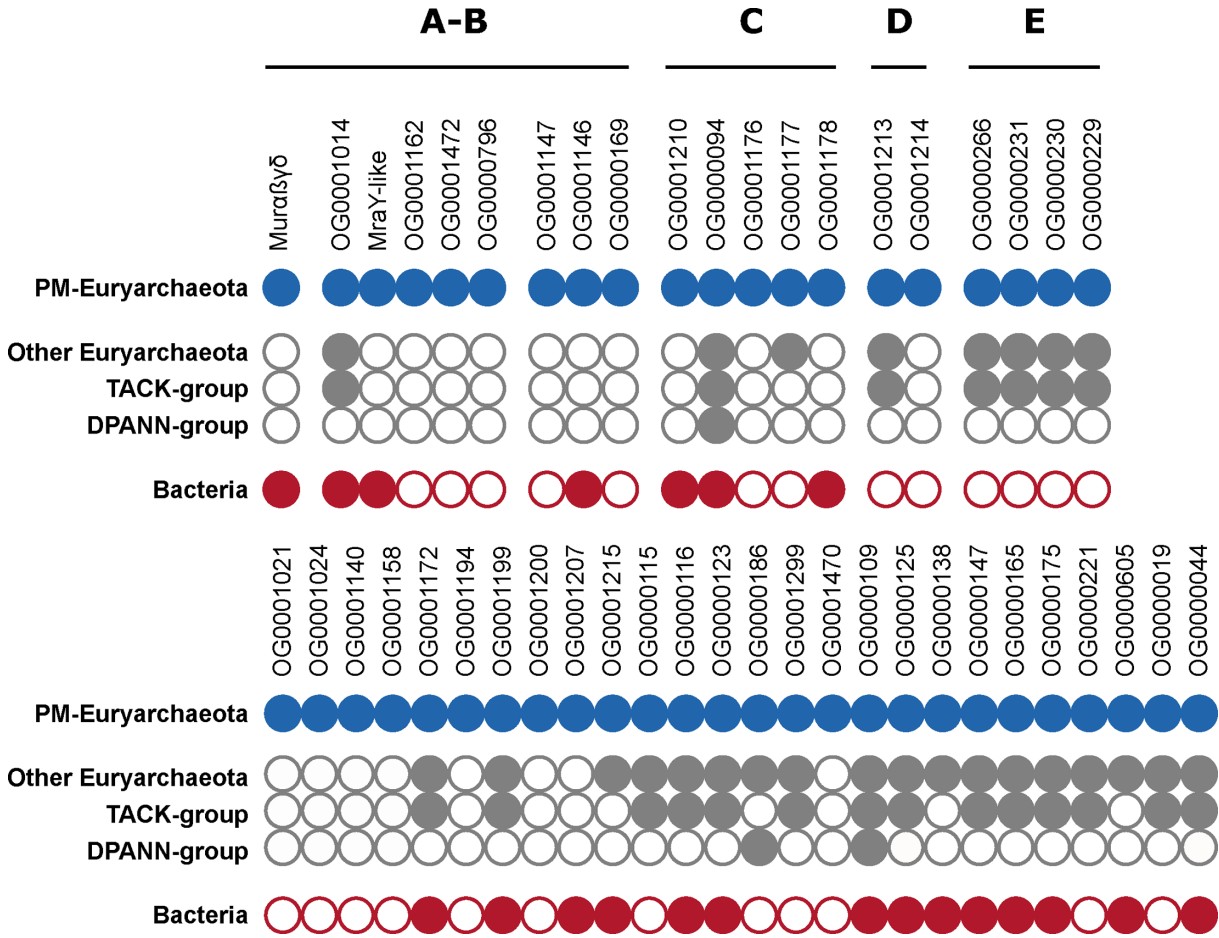

FIG 1  Taxonomic distribution patterns of the 49 retained orthologous groups (OGs). The four OGs OG0001148, OG0001149, OG0001150, and OG0001473 are considered together and referred to as Muraßγδ and OG0001163 as MraY-like. Black lines delineate gene clusters in the genomes of PM-containing archaea (clusters A to E). Full circle = gene present in the taxonomic group; empty circle = gene absent from the taxonomic group.

## MraY-like family

Homology searches revealed that the bacterial homolog of OG0001163 is the glycosyl-transferase 4 (GT4) MraY. According to the NCBI CDD (70), MraY is part of the MraY-like family, which further includes WecA (71), WbpL (72, 73), and eukaryotic and archaeal GPT (74). In addition to the MraY-like OG0001163, our pipeline has highlighted transmembrane proteins in OG0001207 (Fig. 1), for which the only bacterial homolog also has a MraY/WecA-like GT4 domain. Therefore, we decided to add the sequences of OG0001207 to the phylogenetic analysis of the MraY-like family. Although only one sequence similar to OG0001207 had been identified in the bacterial database, 62 additional bacterial OG0001207 homologs were identified in the (larger) prokaryotic database. According to the study of Lupo et al. (75), none of the genomes coding for those protein sequences are considered as contaminated, which suggests that OG0001207 homologs genuinely exist in these bacteria. Overall, a total of 1,267 sequences from the MraY-like family were identified in our databases, corresponding to 1,071 unique sequences. Interestingly, 773 sequences among 1,267 were identified by two or more HMM profiles of the individual members of the MraY-like family. As reference sequences of WecA and WbpL annotated putative sequences from the same monophyletic group, the whole group was considered as WecA/WbpL.

Due to this non-universal taxonomic distribution and lack of an ancestral gene that could be present in the genome of LUCA, the three MraY-like family trees (see Materials

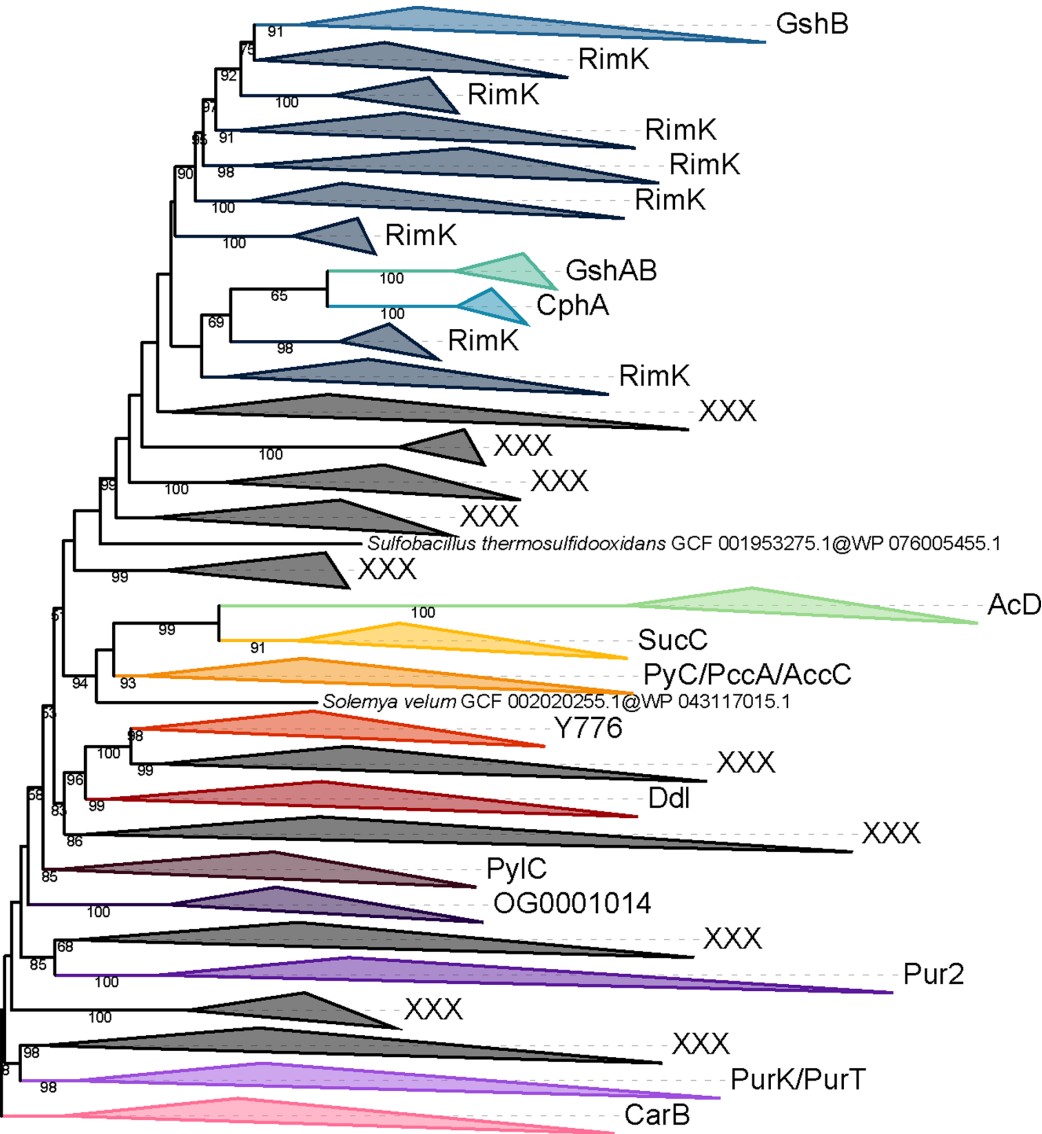

**FIG 2** Phylogenetic tree of the ATP-grasp superfamily rooted on CarB. The tree was inferred from a matrix of 2,194 sequences × 180 unambiguously aligned AAs using IQ-TREE under the LG4X + R4 model. Tree visualization was performed using iTOL. Bootstrap support values are shown if greater or equal to 50%. Branches were collapsed on sequence annotation based on reference sequences. Black collapsed branches correspond to unannotated sequences.

and Methods) were left unrooted. Phylogenetic analysis showed that each of the five members of the MraY-like family is monophyletic and all supported by high bootstrap values. Moreover, MraY formed a clan with WecA/WbpL, while GPT formed a clan with OG0001163 and OG0001207, and these relationships are similar for the three evolutionary models LG4X, C20, and C40 (Fig. 3; Fig. S9 and S10). Taxonomic analysis revealed that MraY and WecA/WbpL are exclusive to bacteria, while GPT is only found in archaea. Regarding OG0001163, it is exclusive to PM-containing archaea, as would be OG0001207, ignoring the few aforementioned exceptions. Together, the clear dichotomy between bacterial and archaeal sequences and the high statistical support (≥90%) at the basal branch of each member of the MraY-like family suggest that LUCA already possessed at least one copy of a *mraY*-like gene.

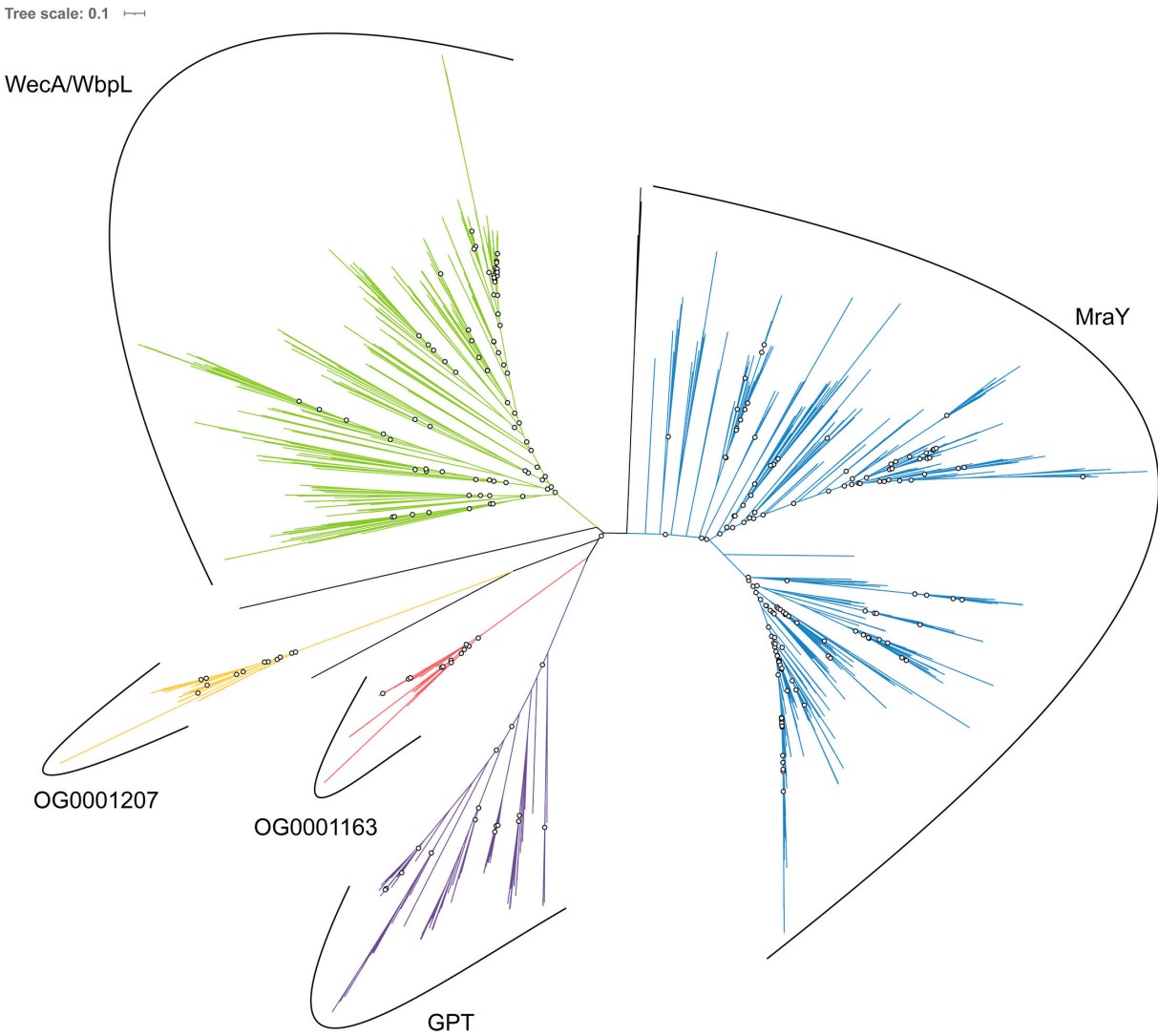

**FIG 3** Unrooted phylogenetic tree of the MraY-like family. The tree was constructed from a matrix of 1,070 sequences × 408 unambiguously aligned AAs using IQ-TREE under the C40 + G4 model. Open circles correspond to bootstrap support values under 90%. Blue sequences correspond to a MraY annotation, green to WecA/WbpL, red to OG0001163 (MraY-like), yellow to OG0001207, purple to GPT, and black to unannotated bacterial sequences.

## Mur domain-containing family

Homology searches allowed us to identify 3,398 unique sequences distributed across 755 prokaryotic organisms. These sequences correspond to 12 members of the Mur domain-containing family, which are the four bacterial MurCDEF, the four archaeal Muraßγδ, MurT, CapB, CphA, and FPGS. Taxonomic distribution within each member protein group revealed that MurCDEF and CphA are specific to bacteria, Muraßγδ are specific to PM-containing archaea, while MurT, CapB, and FPGS are found both in Bacteria and Archaea, albeit not universally. According to the function and ubiquity of FPGS, we assumed that a FPGS protein was already present in LUCA, and trees were rooted on the corresponding clan. The phylogenetic trees, inferred with three models from a matrix of 3,407 sequences × 550 AAs (see Materials and Methods), including the 12 members of the Mur domain-containing family, showed that each member group is monophyletic and supported by high statistical values (bootstrap support around 100; Fig. S11 to S13). In spite of the solid monophyly of each Mur domain-containing family member, the recovered relationships between these members (i.e., the topology of the family tree) appear to depend on the evolutionary model. We made the same observation for

phylogenetic reconstructions based on a smaller matrix, restricted to the most conserved AAs over the full-length alignment (3,386 sequences × 228 AAs) (Fig. S14 to S16).

The relationships between the four bacterial muramyl ligases MurCDEF and their uncharacterized archaeal homologs Muraβγδ were also investigated through phylogenetic analyses using only one out of four potential outgroups among MurT, CapB, CphA, and FPGS, again under the three models (Fig. 4a; Fig. S17 to S27). In these trees, Murα and Murβ always strongly group together (bootstrap support from 86 to 100), and further form a clan with Murγ and MurD in 11 trees out of 12 (support between 81 and 99 in ten trees). Murδ groups with MurC in eight trees, but support is uneven and generally weaker (three trees between 95 and 100 and the five others around 50). Furthermore, MuraβγD and MurδC form a clan in five trees (support between 93 and 98 in four trees), while this larger clan further includes MurT in the three trees where it is present.

To test the robustness of the recovered relationships, we implemented a jackknife approach based on random subsets of sequences representing the 12 members of the Mur domain-containing family. The replicate trees (also built under LG4X, C20, and C40 models) were either combined into consensus trees using ASTRAL (Fig. 4b; Fig. S30 and S31) or automatically searched for clans of interest (see Materials and Methods for details). These resampling analyses confirmed the monophyly of each member of the Mur domain-containing family with the exception of MurT. Indeed, LG4X and C40 ASTRAL trees revealed that MurT sequences can explode into two distinct clans: (i) a large one composed of bacterial and Methanobacteriales sequences and (ii) a smaller one composed of sequences of Methanopyrales and Methanobacteriales, which we termed MurT-like. In addition, ASTRAL trees also confirmed the relationships between the eight muramyl ligases observed in the single-outgroup trees, even if those were blurred by the unstable positions of MurT and MurT-like sequences. Of note, although ASTRAL trees exhibited maximal quartet support values for the relationships between Mur domain-containing family members, the actual jackknife proportions for most clans of interest were much lower (Table S3), which indicates that the precise branching order of the members of the family remains difficult to resolve.

As *murT* and *gatD* genes are clustered in an operon in the genomes of *Staphylococcus* species (32, 76), we performed a homology search of GatD in our local databases. This survey revealed that the Methanobacteriales and bacterial species that harbor a MurT homolog also have a GatD homolog, whereas no GatD homologs are found in archaeal species that only bear MurT-like. The topology of the tree built from the concatenation of MurT and GatD sequences (Fig. 5) showed that the archaeal operon is monophyletic and emerges from Terrabacteria, with *Caldisericum exile* as the closest relative (bootstrap support of 89). In addition, the Terrabacteria sequences, represented by Bacillota (previously named Firmicutes), those of Actinomycetota (Actinobacteria), and a single sequence from Chloroflexota (Chlorofexi), are grossly intermingled. These relationships also apply to all the (larger) trees, including MurT and MurT-like sequences (Fig. 4A and B; Fig. S11 to S19 and Fig. S30 to S33 ),where the MurT-like clan is always basal to the MurT clan.

Overall, our analyses showed that neither MurT nor CphA should be considered as an outgroup for the Mur domain-containing family. Indeed, we observe that MurT sequences form either one or two (MurT + MurT-like) clans, which emerge from the larger clan formed by the six muramyl ligases MuraβγδCD. In spite of the difficulty to determine the exact positions of MurT and MurT-like, both regular trees and resampling analyses tend to indicate that all MurT sequences derive from the same ancestral gene as MuraβγδCD. In contrast to the other members of the Mur domain-containing family, CphA originates from the fusion of two functional domains: (i) an ATP-grasp domain at the N-terminal region (see ATP-grasp superfamily) and (ii) the Mur ligase domain at the C-terminal region. This C-terminal region appears to be closely related to MurE and MurF according to our phylogenetic inferences. Regarding CapB, resampling analyses showed that it is not related to the four bacterial muramyl ligases MurCDEF or to the four archaeal muramyl ligases Muraβγδ, but more likely to FPGS (Table S3), thus indicating that it could

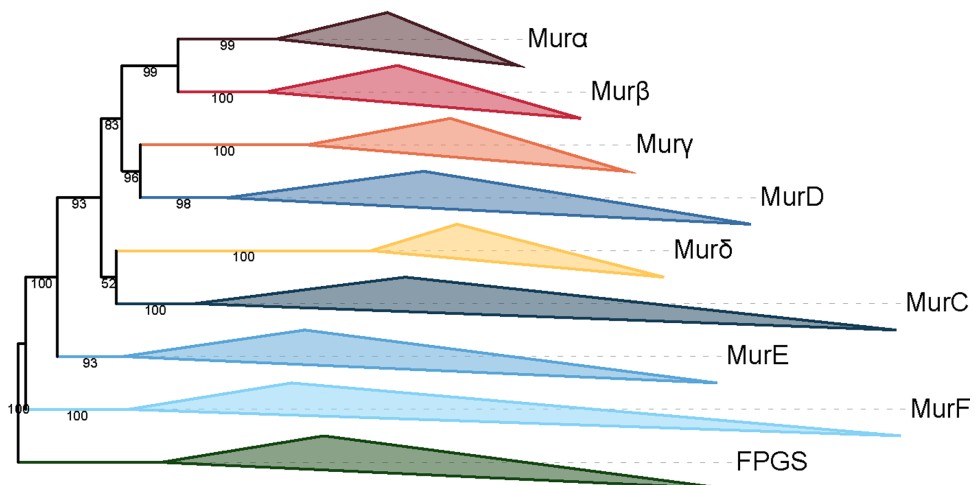

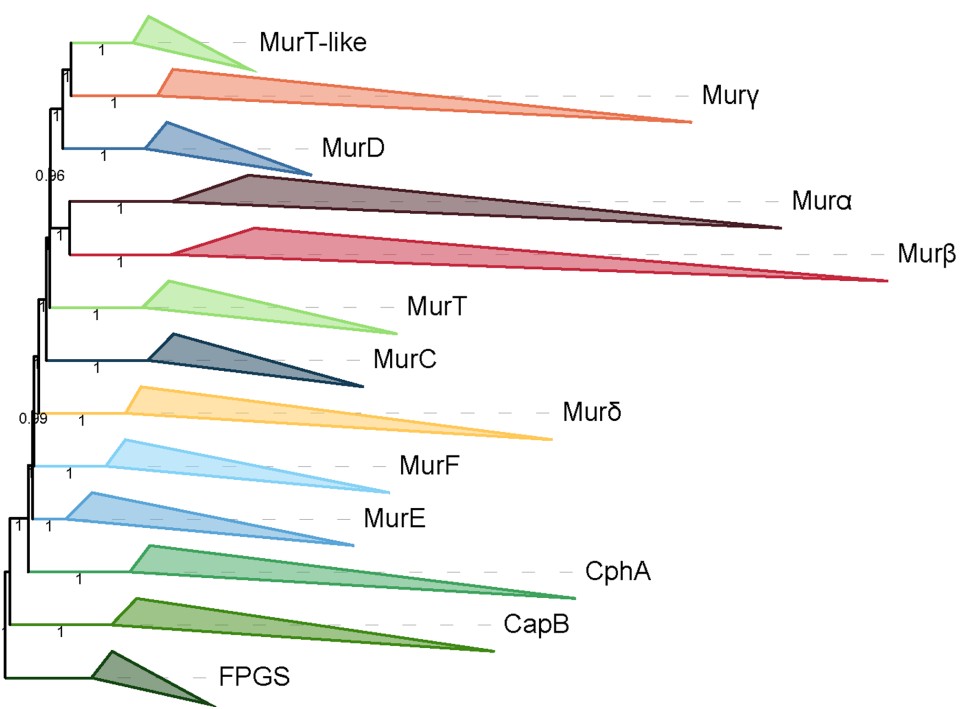

**FIG 4** Phylogenetic trees of the Mur domain-containing family rooted on FPGS. (a) The tree was inferred from a matrix of 3,046 sequences × 543 unambiguously aligned AAs using IQ-TREE under the C40 + G4 model. (b) ASTRAL tree computed from the 1,000 C40 + G4 species resampling trees. Tree visualization was performed using iTOL. Support values are shown if greater or equal to 50% (a: bootstrap proportion; b: quartet support value). Branches were collapsed on identical sequence annotation, and to do so, two divergent sequences, one annotated as a MurE and the other as a MurF, were considered as CphA.

be used as an outgroup to study the relationships between MurCDEF and Muraßγδ. However, unlike FPGS, the taxonomic distribution of CapB is more restricted, being found only in Gammaproteobacteria, Bacilli, Synergistota (Synergistetes), Halobacteria and a

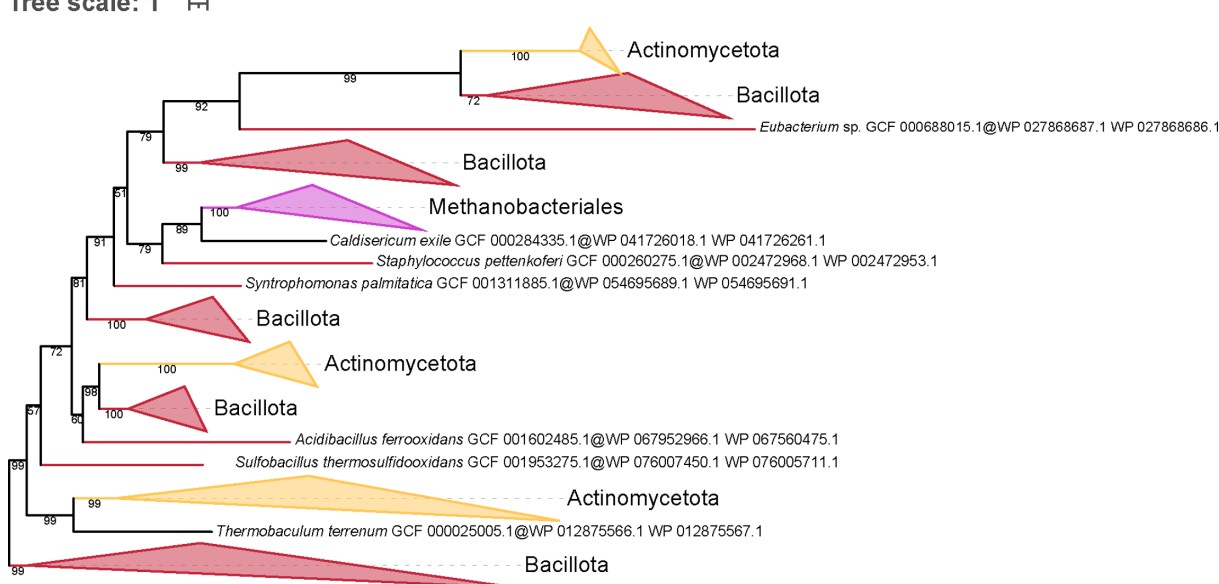

**FIG 5** Phylogenetic tree of the concatenation of MurT/GatD rooted on the largest Bacillota clan. The tree was inferred from a matrix of 229 sequences × 653 unambiguously aligned AAs using IQ-TREE under the C40 + G4 model. Tree visualization was performed using iTOL. Bootstrap support values are shown if greater or equal to 50%. Branches were collapsed based on taxonomic clans.

few Methanosarcinales, and Korarchaota, which eventually makes it less suitable to the task than FPGS.

## DISCUSSION

### Improved catalog of pseudomurein biosynthetic genes

In this study, we devised a pangenomic strategy to identify all the genes potentially involved in PM biosynthesis in Archaea. In contrast to Subedi et al. (54), who relied on similarity searches to identify candidate genes, our protocol consisted in clustering protein sequences from the whole proteomes of five PM-containing archaea and five non-PM Euryarchaeota into OGs, followed by a semi-automated analysis of the taxonomic distribution of the resulting OGs. This strategy enabled the identification of a total of 49 OGs, among which some are exclusively specific to PM-containing archaea, while others contain paralogous genes that are specific to those archaea. Only a few of them could be associated with pseudomurein biosynthesis and correspond to the two syntenic regions described in (54). With respect to the latter study, two more genes were detected due to their association with either cluster A or B in some lineages: OG0000766, a polytopic transmembrane protein, and OG0000169, an extracellular protein. Both have an unknown function but are specific to PM-containing archaea. In addition to the two clusters, the similarity searches of (54) also identified eight more genes in PM-containing archaea: *glmS*, *glmM*, *glmU*, *galE*, *murG*, *uppS*, *uppP*, and a flippase. However, our own approach did not recover these genes, an expected outcome that can be explained by two reasons: (i) genes like *glmS*, *glmM*, *glmU*, or *galE* are nearly ubiquitous among prokaryotes (54) and have no specific paralogs in PM-containing archaea, (ii) genes, like *murG or uppS*, are missing in some genomes of the five PM-containing archaea and; therefore, their role in PM biosynthesis is questionable. Our OG0000044 includes the transglutaminase genes mentioned in Leahy et al. (52), which are enzymes that catalyze the bond formation between L-Glu and L-Lys residues in proteins (77), but the corresponding genes are not associated with any cluster. Yet, *in silico* predictions show that these enzymes are exported toward the extracellular space. Therefore, OG0000044 is the best candidate to perform a bond formation between L-Glu in position 5 of one chain to L-Lys in position 3 of another chain during PM cross-linking. In comparison to the study

of Subedi et al. (54), we have additionally identified five OGs neither associated with cluster A or B: OG0001024, OG0001140, OG0001158, OG0001194, and OG0001470. These OGs do not exhibit any known function but are exclusive to PM-containing archaea (Fig. 1). Given their taxonomic distribution, they might play a role in PM biosynthesis, although further investigation is required to confirm this hypothesis.

## Origin and evolution of Mur ligases

Our phylogenetic analyses of the Mur domain-containing family show that each member of the family is monophyletic. However, the relationships between those members are difficult to resolve owing to the low remaining phylogenetic signal, and because phylogenetic artifacts, such as long-branch attraction, probably affect phylogenetic reconstruction, especially in trees including all non-Mur "outgroups" (78). Indeed, compared with MurCDEF, archaeal muramyl ligases (here termed Muraßγδ) are characterized by very long branches; for example, Murδ has experienced more than one substitution per site since its probable separation from MurC. When focusing on Mur trees with only one outgroup, the topology is quite robust to the use of different evolutionary models, as it is in species resampling analyses featuring all outgroups (Table S3). In this topology, MurD forms a clan with Murα+Murß+Murγ, MurC a clan with Murδ, and MurE a clan with MurF, and all these relationships are also recovered in unrooted trees devoid of any outgroup (Fig. S34 to S36). Considering a two-domain tree of life, our results enable the exclusion of potential evolutionary scenarios, in which LUCA would have possessed either one, two, or four *mur* genes vertically inherited by both Bacteria and Archaea (Fig. 6A through C). The first scenario implies two series of three gene duplications, independently in Bacteria and Archaea. If true, this scenario would be evidenced by a bipartition between all bacterial sequences and all archaeal sequences (Fig. 6A). The second scenario posits a duplication event in LUCA, followed by additional gene duplications in bacterial and archaeal lineages, resulting in two mirror subtrees, each bearing two archaeal and two bacterial genes (Fig. 6B). In the third scenario, where LUCA already possessed four *mur* genes, each archaeal gene would have been associated with its bacterial counterpart (i.e., ortholog) in the tree (Fig. 6C). Besides, two potential scenarios where archaeal genes would have been acquired by horizontal transfer from bacteria can be ruled out (Fig. 6C and D). Indeed, the topology of Fig. 6C would also be recovered after the transfers of the four bacterial genes to archaea, whereas a scenario involving the single transfer of any form of bacterial gene to the ancestor of PM-containing archaea, followed by three duplications, would yield a tree where the four archaeal Mur ligases emerge from bacterial Mur ligases (Fig. 6D). Finally, it is worth noting that, in an unrooted phylogeny, the last scenario of Fig. 6D would be indistinguishable from the first one depicted in Fig. 6A.

As previously stated, early analyses of their biosynthetic pathways have suggested that neither PG nor PM was a feature of LUCA (4, 11, 54, 55). Therefore, in a two-domain tree of life, LUCA probably did not possess the various muramyl ligases presently involved in cell-wall biosynthesis. Furthermore, since the LBCA already possessed a complete *dcw* gene cluster (28), and given that PM is restricted to Methanopyrales and Methanobacteriales (2), we propose a scenario for Mur ligase evolution (Fig. 6E). In this scenario, the ancestral gene of *murCDEF* was duplicated a first time in the pre-LBCA lineage to yield the ancestral genes of *murCD* and *murEF*, followed by a second round of duplications, which led to the current four bacterial muramyl ligases. Some evidence indicates that the duplication of the *murEF* ancestral gene to yield *murE* and *murF* could have occurred later than the duplication of the *murCD* ancestral gene. In fact, *murE* and *murF* genes are always in tandem in the *dcw* cluster of most bacterial species, as well as in the reconstruction of the LBCA *dcw* cluster (28), and can even be expressed as a single fusion protein MurE–MurF (79). Moreover, in the majority of our Mur domain-containing family trees, MurE and MurF have slightly shorter branches than those of MurC and MurD. As we rule out the scenarios involving one and four HGTs for the origin of archaeal muramyl ligases (Fig. 6C and D), those must have stemmed from either two or three HGT

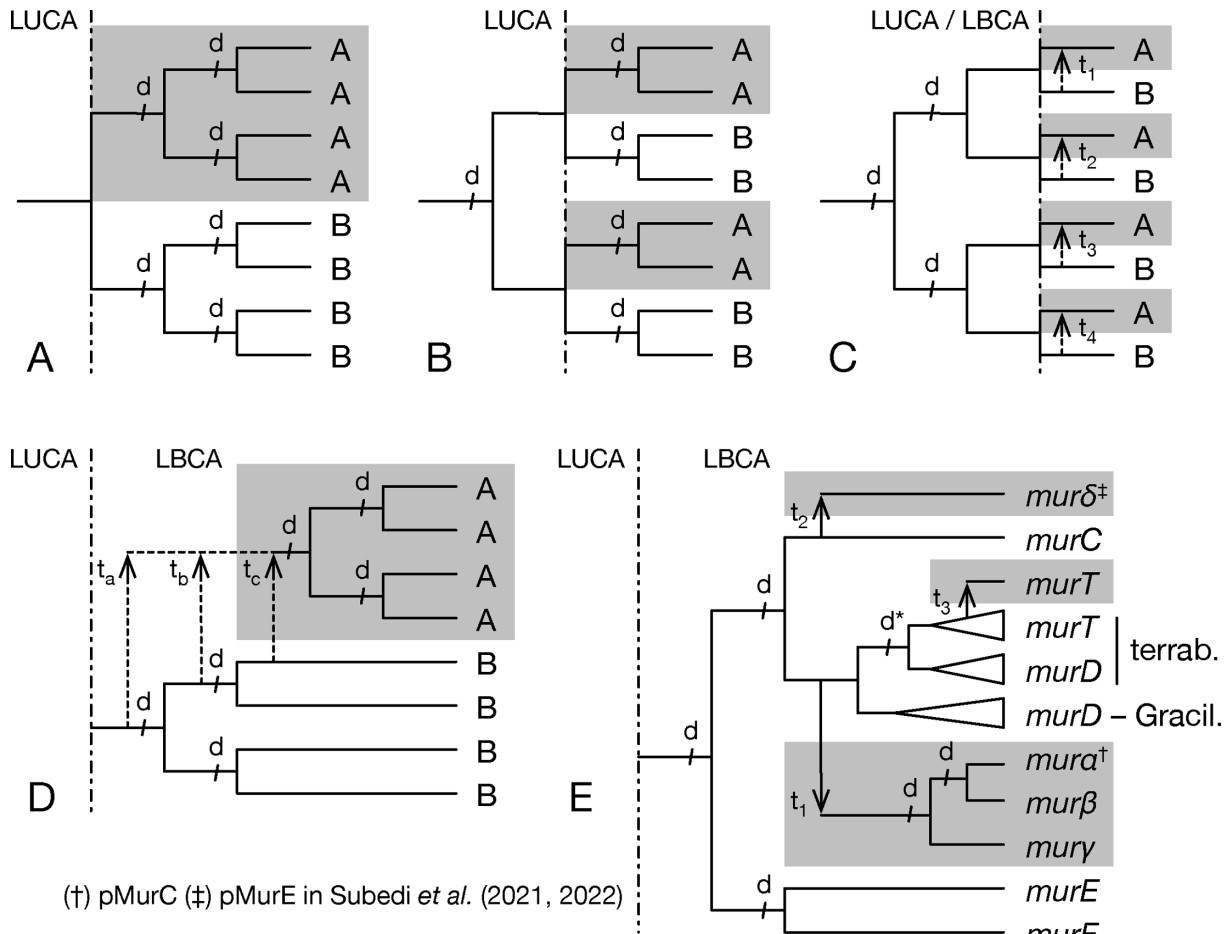

**FIG 6** Evolutionary scenarios for the origins of the main Mur-domain containing genes in Archaea and Bacteria. Panels (A-D) were theoretically possible but are ruled out by the phylogenetic analyses performed in this study, whereas panel (E) is the most precise hypothesis that can be proposed in the current state of knowledge. (A) LUCA possessed a single *mur* gene vertically inherited by both Archaea and bacteria, and then independently duplicated (events d) thrice in each domain. (B) LUCA was already carrying two paralogous *mur* genes, each one experiencing an additional duplication after their vertical transmission to both Archaea and Bacteria. (C) All four *mur* genes were already existing in LUCA and vertically transmitted to both Archaea and Bacteria. Alternatively, these four *mur* genes were only present in the LBCA and were each horizontally transferred (events $t_1$ to $t_4$) to (PM-containing) Archaea. (D) The LBCA possessed the four *mur* genes, and any of these (or even one of their ancestors) was horizontally transferred (event $t_a$, $t_b$ or $t_c$) to (PM-containing) Archaea, where it experienced three duplications. (E) As in (D), the LBCA possessed the four *mur* genes, but only two of them were horizontally transferred (events $t_1$ and $t_2$) to (PM-containing) Archaea, in which one was duplicated twice. In detail, *murC* gave rise to *murδ*, whereas *murD* led to *muraβγ*. Moreover, an independant duplication (d\*) of *murD* in Terrabacteria yielded *murT*, which was then horizontally transferred to (PM-containing) Archaea. († ) *mura* is called pMurC in Subedi et al. (54) (‡ ) *murδ* is called pMurE in Subedi et al. (56).

events. In fact, our phylogenetic analyses consistently suggest only two instances of HGT. After the diversification of the LBCA, the *murD* gene was transferred to the common ancestor of PM-containing archaea, then *murD* experienced two duplications that yielded *mura*, *murß* and *murγ* (our nomenclature). There is strong evidence that *mura* and *murß* arose from a gene duplication: the corresponding proteins group together in almost all phylogenetic reconstructions (in both rooted and unrooted trees) and, as for MurF and MurE, their genes are in tandem in the genome of most PM-containing archaea, while some Methanobrevibacter and Methanothermobacter genomes (two genera of Methanobacteriales) code for a Mura-Murß fusion protein (54). As for the first, older, duplication of *murD*, leading to *murγ* and the *muraß* ancestor, it is visible in unrooted trees, where *mura*, *murß*, and *murγ* form a clan. Regarding *murδ*, the phylogenetic analyses mostly suggest a *murC* affinity, and thus an independent gene transfer event from bacteria.

Compared with other members of the Mur domain-containing family, the four bacterial and the four archaeal muramyl ligases are the only enzymes featuring three homologous structural domains (N-terminal, central, and C-terminal) (80). In bacterial sequences, the N-terminal domain binds to PG precursors, the central domain binds ATP, while the C-terminal binds the AA that is transferred to the nascent pentapeptide. The N-terminal domain exhibits the highest sequence and structural diversity, whereas the central domain is the most conserved (80). In addition, the accepted dichotomy of MurC/MurD versus MurE/MurF is only based on sequence similarities and 3D structures of the N-terminal domain (80, 81). Thus, the observed varying level of conservation of the different domains probably results in a phylogenetic signal that is not uniform along the sequences. This may explain the inconsistency between the study of Subedi et al. (56) and ours, where they associate Murδ with MurE/F, based on the structure of the N-terminal domain, whereas our phylogenetic analyses of the full-length sequences rather indicate an affinity to MurC. It is noteworthy that, in the CapB and CphA outgroup trees computed with the C40 model (Fig. S22 to S25), Murδ exceptionally branches inside the MurE clan, within Bacillota. Even though such an alternative relationship would fit the conclusion of Subedi et al. (56), the analysis of the two matrices under the more sophisticated PMSF LG + C60+G4 model (Fig. S28 and S29) did not yield that latter topology, and instead supported the first solution.

Although there are several clues hinting to gene transfers of muramyl ligases from bacteria to PM-containing archaea, the archaeal sequences (Murαßγδ) almost never branch within bacterial muramyl ligase sequences (MurCDEF) in our phylogenetic trees, and these trees do not provide indications about the direction of the transfers. However, the apparent monophyly of each bacterial member of the family could also be an artifact due to fast-evolving sequences in archaeal species. This kind of artifact has already been reported, e.g., with plastidial genes in eukaryotes, which rarely branch within (and rather sister to) Cyanobacteria (82), although the endosymbiotic origin of the plastid is widely accepted (83). Nevertheless, we detected a clear transfer from bacteria to PM-containing archaea, involving the *murT* gene. Previously, MurT had been described in *Staphylococcus* spp.*, Streptococcus pneumoniae*, and *Mycobacterium tuberculosis* (32, 33, 76, 84). Here, our analyses revealed that MurT is not ubiquitous in Bacteria, being only found in Bacillota, Actinomycetota, *Caldisericum exile* (Caldisericota), and *Thermobaculum terrenum* (Chloroflexota). We also identified *murT* homologs in Archaea, specifically in Methanopyrales and Methanobacteriales. Interestingly, the bacteria that harbor MurT mostly have one copy of the *murT* gene, while some PM-containing archaea have two copies, which we named *murT* and *murT-like*. Hence, Methanobacteriales can possess only MurT or only MurT-like or both, whereas the few available Methanopyrales solely have one MurT-like gene. Archaeal MurT sequences clearly group with the sequence of *C. exile*, which further branches within Terrabacteria. The tree of the MurT partner, GatD, is similar, and this topology is confirmed by the concatenation of MurT and GatD sequences. This suggests that both genes were transferred together from a Terrabacteria lineage, with *C. exile* as the closest extant relative of the donor lineage. Establishing the exact phylogenetic position of *C. exile* in the bacterial diversity remains challenging due to the limited genomic data available for Caldisericota. In 2022, Léonard et al. showed that *C. exile* may lie between Terrabacteria and Gracilicutes, which means that it could be affiliated to one or the other superphylum, depending on the rooting of the bacterial tree. The taxonomic distribution MurT and GatD among bacteria suggests that *C. exile* might belong to Terrabacteria, and a recent study reinforces this view by positioning Caldisericota within Terrabacteria, but still based on the same single genome (85). According to the taxonomic distribution of archaeal MurT/GatD and Murαßγδ, we hypothesize that the gene transfers of *murT/gatD* and the two ancestor genes of *muraßγδ* both occurred before the diversification of PM-containing archaea. In contrast, the origin of the *murT-like* gene is enigmatic, owing to its position basal to the MurT clan, even though one possible explanation would be a duplication of *murT* in the last

common ancestor of Methanopyrales and Methanobacteriales, followed by differential losses of either *murT/gatD* or *murT-like* in descendant lineages.

## Uncompelling evidence for Ddl and MraY orthologues in archaea

In cluster A, there are two adjacent genes, OG0001014 and OG0001163, that show a homology with genes of the bacterial *dcw* gene cluster, respectively, *ddl* and *mraY*. Both genes are exclusive to Methanopyrales and Methanobacteriales species. Thus, it has been proposed that their products are probably involved in PM biosynthesis (54). Having concluded that archaeal muramyl genes originate from gene transfers from an ancient bacterial lineage, we further investigated whether OG0001014 and OG0001163 genes were also transferred from bacteria. However, our extensive phylogenetic analyses of the ATP-grasp superfamily and the MarY-like family revealed that neither OG0001014 nor OG0001163 is specifically associated with Dld or MraY. Indeed, although trees of the ATP-grasp domain show members of the superfamily that are mostly monophyletic and supported by high statistical values, the position of Ddl and OG0001014 in the tree is not stable across our seven trees. In addition, OG0001014 never clusters with Ddl (except in the C20 tree; support value of 63) but always features a long branch, which can explain the difficulty to position it accurately (i.e., long-branch attraction artifact). In 2012, Popa et al. (59) experimentally characterized OG0001014 and concluded that it was the smallest CPS closely related to the large subunit of a "true" CPS, CarB. However, OG0001014 never groups with CarB in any of our trees, while its genetic environment suggests that it is involved in PM biosynthesis. For their part, Subedi et al. (54) proposed that OG0001014 could be involved in the three-step activation of L-Glu to UDP-N-glu-tamyl-γ-phosphate (UDP-N-Glu-γ-P). Based on these ideas and our own observations, we propose that the reported CPS function of this enzyme might not be correct and that the kinase activity measured by Popa et al. actually plays a role in the addition of a phosphate to L-Glu, either in the first step to form $N^{\alpha}$-P-Glu or in the third step to form UDP-$N^{\alpha}$-Glu-γ-P (9).

Regarding the phylogenetic analyses of the MraY-like family, the archaeal MraY-like does not appear to have evolved from the bacterial MraY. Indeed, bacterial and archaeal proteins are clearly separated in all unrooted phylogenetic trees, although archaeal monophyletic groups are characterized by long branches, especially OG0001207, which could lead to strong phylogenetic artifacts. In contrast to Mur domain-containing family and ATP-grasp superfamily trees, MraY-like family trees were left unrooted. In fact, none of the MraY-like family members is found in both Bacteria and Archaea. As shown in the Results section, MraY and WecA/WbpL are only present in bacterial species, GTP is ubiquitous in archaea, while MraY-like and OG0001207 are exclusive to PM-containing archaea. In addition, WecA/WbpL is the only monophyletic group where some organisms bear two sequences, which could indicate that WecA and WbpL are two paralogs. According to the taxonomic distribution of MraY, WecA/WbpL and GPT, we propose a scenario where an ancestral GT4 gene found in LUCA was vertically transmitted to both Archaea (GPT) and Bacteria (the ancestral gene coding for both MraY and WecA/WbpL). The bacterial gene was then duplicated once to yield *mraY* and *wecA/wbpL*. Thus, GPT would be the ortholog of MraY and WecA/WbpL, while MraY and WecA/WbpL would be paralogous. For this phylogenetic analysis of the MraY-like family, we followed the family as defined in the NCBI CDD (https://www.ncbi.nlm.nih.gov/Structure/cdd/cddsrv.cgi?uid=264002) to increase the sequence sampling. In theory, it is possible that we undersampled the family. Indeed, the GT4 domain is also present in other proteins, like MurG (79, 86), which are not part of the MraY-like family. A proper way to study the origin of the MraY-like family would be to infer a phylogenetic tree of the GT4 domain (as we did for the ATP-grasp domain). However, such an analysis would be very time-con-suming due to the large number of GT sequences (87). For now, overlapping HMM search results starting from the different family members do not suggest any undersampling issue. Actually, it could be the opposite, with our tree including sequences that are not

part of the MraY-like family, such as the bacterial homologs of OG0001207, which have a very long branch in spite of featuring a MraY/WecA-like GT4 domain.

## Nature of the ancestral cell wall

The current architecture of PG and PM is well-known, but our work indicates that both polymers were different in their early evolutionary state, i.e., before acquisition and diversification of their respective muramyl ligases. However, inferring the ancestral state of either PG or PM is almost impossible because those evolved in the stem branch of Bacteria or class I methanogen Archaea (including both Methanobacteriales and Methanopyrales), respectively, before the last common ancestors of extant organisms. As other Mur-domain containing proteins, like CapB, FPGS, MurT, or CphA, bind amino acids with an α-carboxylic acid group (i.e., aspartic acid and glutamic acid), we speculate that the first muramyl ligase proteins were also associated with those AAs. Moreover, glutamic acid is one of the most abundant AAs in many organisms, and it participates in a wide array of metabolisms (88). Therefore, glutamic acid could be one of the first AAs to have been selected by muramyl ligases. In *Bacillus*, the complex formed by CapB, CapC, CapA, and CapE recruits L-Glu or D-Glu to synthesize the poly-γ-glutamic acid capsule. This kind of cell wall has also been suggested to occur in *Haloquadratum walsbyi*, based on genomic analyses. *H. walsbyi* is classified in Halobacteria, a class of Euryarchaeota characterized by a diverse variety of cell walls: S-layer, sulfated heteropolysaccharides, halomucin and glutaminylglycan. The latter is composed of poly-γ-L-glutamate, to which are linked two types of oligosaccharides (2). Analyses showed that CapB is ubiquitous in Halobacteria, indicating that CapB could be involved in glutaminylglycan biosynthesis. Consequently, we suggest that this simpler cell wall could resemble the ancient forms of PG and/or PM.

## MATERIALS AND METHODS

### Protein sequence databases

Three local mirrors of NCBI RefSeq were used during this study: (i) an archaeal database composed of the 819 whole genomes that were available on 7 March 2019, (ii) a bacterial database of 598 representative genomes selected by the ToRQuEMaDA pipeline (89), and (iii) a prokaryotic database of 80,490 genomes, already used in (90). To assemble the bacterial database, ToRQuEMaDA was run in June 2018, according to a "direct" strategy and using the following parameters: dist-metric set to JI (Jaccard Index), dist-threshold set to 0.86, clustering-mode set to "loose," and pack size set to 200.

### Identification of candidate proteins for pseudomurein biosynthesis

Protein orthologous groups (OGs) were built from the conceptual translations of 10 archaeal whole genomes using OrthoFinder v2.2.1 (91) with default parameters. These archaeal genomes correspond to five organisms experimentally shown to have pseudomurein (PM) (GCF_000008645.1 [92], GCF_000016525.1 [93], GCF_000166095.1 [94], GCF_002201915.1 [95], and GCF_900095295.1 [96]) and five without PM (GCF_000011185.1, GCF_000013445.1, GCF_000017165.1, GCF_000025285.1, and GCF_000251105.1) and were downloaded from the NCBI RefSeq database on 7 March 2019. Then, taxonomic filters were applied to the OGs using classify-ali.pl v0.212670 in order to select candidate proteins for PM biosyntesis. Hence, we first looked for OGs with protein sequences from all five PM-containing archaea or from one Methanopyrales and three Methanobacteriales or from four Methanobacteriales. To identify OGs corresponding to a widespread gene that would also include a paralog potentially specific to PM-containing archaea, we used the same taxonomic criteria but set the "min_copy_mean" option to 1.75 for PM-containing archaea and to 1.25 for other species (see YAML configuration files for details). In addition, three HMM profiles from NCBI Conserved Domain Database (CDD) (70) featuring "pseudomurein"

in their annotation were downloaded on 18 December 2020. Then, the profiles were used to identify homologs in the conceptual translations of the five PM-containing archaea with hmmsearch from the HMMER package v3.3 (97) with default parameters. Matching protein sequences were graphically selected using the Ompa-Pa v0.211430 interactive software package (A. Bertrand and D. Baurain; https://metacpan.org/dist/Bio-MUST-Apps-OmpaPa) with the "max_copy" option set to 20 and "min_cov" to 0.7. Finally, the corresponding OGs were added to the selection.

## Genetic environment analysis of candidate proteins and *in silico* characterization of their domains

Genetic environment databases were built for the genes of the selected OGs using the "3 in 1" module of GeneSpy (98). Functional domains were predicted using InterProScan v5.37–76.0 (99), along with SignalP v5.0b (100)and TMHMM v2.0c (101). InterProScan was used with default parameters and we disabled the precalculated match lookup, while the SignalP organism option was set to "arch." To avoid misprediction by TMHMM, the signal peptide was first removed from the original sequences when the cleavage site prediction probability was greater than or equal to 0.1.

## Filtering of candidate proteins

To rescue potential pseudogenes or mistranslated proteins missing in selected OGs with protein sequences from only four (out of five) PM-containing archaea, Forty-Two v0.213470 was run in TBLASTN mode on the whole genomic sequences of the five PM-containing archaea. Then, classify-ali.pl was used again to retain only the OGs having sequences from all five PM-containing archaea. To enrich OGs with further archaeal orthologs, a second round of forty-two.pl in BLASTP mode was performed using the archaeal database of 819 whole genomes (see YAML configuration files for details). Each enriched OG was aligned using MAFFT L-INS-i v7.273 (102). From those alignments, HMM profiles were built using the HMMER package, and bacterial homologs were identified separately in the bacterial and the prokaryotic databases. Protein sequences were graphically selected using Ompa-Pa with "max_copy" and "min_cov" options set to 20 and 0.7, respectively. For each OG, identical length and e-value thresholds were used for both databases when selecting homologous proteins.

## Phylogenetic analyses

### ATP-grasp superfamily

In order to select a set of representative sequences containing the ATP-grasp domain, we first built a HMM profile from the alignment of the OG containing archaeal ATP-grasp domain proteins using the HMMER package. This profile was uploaded to the HMMER website (https://www.ebi.ac.uk/Tools/hmmer/search/hmmsearch) from which we retrieved homologous sequences (excluding eukaryotes) from the Swiss-Prot database (103). From those sequences, homologs were identified in our local bacterial databases using the HMM profile and Ompa-Pa. In parallel, the archaeal OGs (see "Collection of candidate proteins for pseudomurein biosynthesis," above) homologous to the Swiss-Prot proteins were identified using NCBI BLASTp v2.2.28+ (104) and enriched using Forty-Two with the archaeal database as "bank." Finally, all archaeal and bacterial homologous sequences were merged into one single file. To identify most ATP-grasp-containing domain proteins in our local databases, the merged file was aligned using MAFFT L-INS-i, and the alignment was masked using the mask-ali.pl perl script (D. Baurain; https://metacpan.org/dist/Bio-MUST-Core) to isolate the ATP-grasp domain. From this domain alignment, an HMM profile was built using the HMMER package to identify ATP-grasp domain-containing homologs in our archaeal and bacterial databases, and homologous sequences were selected using Ompa-Pa. Protein sequences with two ATP-grasp domains (i.e., CarB) were cut at half-length, then both complete and half-sequences were aligned using MAFFT and their ATP-grasp domain

again isolated using mask-ali.pl. Protein sequences were deduplicated using cdhit-tax-fil-ter.pl perl script (V. Lupo and D. Baurain; https://metacpan.org/dist/Bio-MUST-Drivers) with the "keep-all" option enabled and the identity threshold set to 0.65, then tagged using a BLAST-based annotation script (part of Bio-MUST-Drivers), and highly divergent sequences were removed using prune-outliers.pl v0.213470 with the "evalue" option set to 1e−3, "min-hits" to 1, "min_ident" to 0.01, and "max_ident" to 0.2. Finally, sequences were realigned with MAFFT L-INS-i. Conserved sites were selected using ali2phylip.pl v0.212670 (D. Baurain; https://metacpan.org/dist/Bio-MUST-Core) with the "min" and "max" options set to 0.3. The resulting matrix of 2,194 sequences × 180 AAs was used to infer phylogenetic domain trees using IQ-TREE v1.6.12 (105) with 1,000 ultrafast bootstrap (UFBoot) replicates (106) and under four models: LG4X + R4, C20 + G4, C40 + G4, and PMSF LG + C60+ G4. In total, seven trees were computed because we tested the effect of increasing the number of iterations from 1,000 to 3,000 for the C20 and C40 models and from 3,000 to 5,000 for the PMSF model.

### MraY-like family

The two OGs (see Identification of candidate proteins for pseudomurein biosynthe-sis) containing proteins predicted with a domain glycosyltransferase 4 were enriched in bacterial homologs using Forty-Two in BLASTP mode. In parallel, representative sequences from other members of the MraY-like family (https://www.ncbi.nlm.nih.gov/Structure/cdd/cddsrv.cgi?uid=264002) were downloaded from the UniProtKB (107) database: WecA (P0AC78, P0AC80, and Q8Z38), GPT (P96000 and B5IDH8), and WbpL (G3XD50 and A0A379IBB8). The three files were then enriched in bacterial and archaeal (if any) homologs using Forty-Two. Finally, the five files were aligned using MAFFT L-INS-i.

To better explore the diversity of the MraY-like family, HMM profiles were built from those alignments, and homologous sequences were selected from HMMER hits on the bacterial database using Ompa-Pa. All homologous protein sequences were merged into one file and tagged using a BLAST-based annotation script (part of Bio-MUST-Drivers) and aligned using MAFFT L-INS-i. Conserved sites were selected using ali2phylip.pl with the "min" and "max" options set to 0.2. A first guide tree was computed from the resulting matrix of 1,070 sequences × 410 AAs using IQ-TREE with 1,000 UFBoot under the LG4X + R4 model. From this guide tree and automated annotation, all sequences were manually tagged using "treeplot" from the MUST software package (108). According to their annotation, protein sequences of each member of the MraY-like family were aligned using MAFFT L-INS-i, then all members were realigned using Two-Scalp v0.211710 (A. Bertrand, V. Lupo and D. Baurain; https://metacpan.org/dist/Bio-MUST-Apps-TwoScalp) with the "linsi" option enabled. Finally, ali2phylip.pl was used to select conserved sites with the "min" and "max" options set to 0.2, and the resulting matrix of 1,070 sequences × 408 AAs was used to infer phylogenetic trees with IQ-TREE under three models (i.e., LG4X + R4, C20 + G4, C40 + G4) and 1000 UFBoot.

### Mur domain-containing family

After enrichment of the OGs with archaeal and bacterial homologs, the multiple OGs corresponding to the Mur domain-containing family were merged into one single (unaligned) file. In parallel, reference protein sequences from additional members of the Mur domain-containing family were downloaded into three separate files using the command-line version of the "efetch" tool v10.4 from the NCBI Entrez Programming Utilities (E-utilities): CapB (P96736), MurT (Q8DNZ9, A0A0H3JUU7, A0A0H2WZQ7), and CphA (P56947, O86109, and P58572). Forty-Two in BLASTP mode was run, in two rounds, on the four files, using both bacterial and archaeal databases as "bank," in a final effort to sample the diversity of Mur domain-containing proteins. Then, fusion proteins were cut between the two protein domains, and half-sequences with no Mur ligase domain were discarded. The enriched files were merged, and protein sequences were deduplicated using the cdhit-tax-filter.pl with the "keep-all" option enabled, and the identity threshold set to 1. Mur domain-containing family proteins were tagged using a BLAST-based

annotation script (part of Bio-MUST-Drivers) with an e-value threshold of 1e−20. Protein sequences were aligned using MAFFT (default mode), and conserved sites were selected using ali2phylip.pl with the "max" option set to 0.3. A first guide tree was computed with IQ-TREE under the LG4X + R4 model with 1,000 UFBoot. Based on the automatic annotation, all protein sequences were manually tagged following the guide tree using "treeplot" from the MUST software package.

In order to improve phylogenetic analysis, the alignment of the Mur domain-containing family was refined as follows: (i) sequences from the different members of the family were exported to distinct files and aligned using MAFFT L-INS-i, (ii) using the "ed" program from the MUST software package, misaligned sequences were manually transferred to a ".non" file, and then, reduced files were realigned using MAFFT L-INS-i, (iii) realigned files and ".non" files were merged, and all sequences were aligned using Two-Scalp with the "linsi" and "keep-length" options enabled. Conserved sites were selected using ali2phylip.pl with the "max" and "min" option set to 0.3. Phylogenetic analysis was performed on the resulting matrix of 3,407 sequences × 550 AAs using IQ-TREE with 1,000 UFBoot under three models of sequence evolution: LG4X + R4, C20 + G4, and C40 + G4.

The concatenated MurT/GatD tree was constructed as follows: first, three sequences of GatD (Q8DNZ8, A0A0H3JN63, and A0A0H2WZ38) were downloaded using "efetch." Then, two rounds of Forty-Two in BLASTP mode were run, using both bacterial and archaeal local databases. In order to identify the orthologs of GatD, a preliminary tree was constructed using IQ-TREE with 1,000 UFBoot under the LG4X + R4 model (data not shown). The non-deduplicated sequences (see above) identified as orthologs of MurT and GatD were concatenated and aligned using MAFFT with "auto" and "reorder" options enabled. Finally, conserved sites were selected using ali2phylip.pl with the "max" option set to 0.8, and the resulting matrix of 229 sequences × 640 AAs was used to infer phylogenetic trees with IQ-TREE under the three usual models (i.e., LG4X + R4, C20 + G4, C40 + G4) with 1,000 UFBoot.

The jackknife.pl perl script (part of Bio-MUST-Drivers) was used for species resampling analysis with the "linsi" option enabled, "min" and "max" set to 0.3 and "n-process" to 1,000. The 1,000 resulting alignments were used to infer phylogenetic trees using IQ-TREE with 1,000 UFBoot under the LG4X + R4, C20 + G4, and C40 + G4 models. Clan support values were assessed using the parse_consense_out.pl perl script (109) with the "mode" option set to "tree." Consensus trees were computed from the 1,000 replicate trees using ASTRAL v5.7.7 (110) with default options.

## ACKNOWLEDGMENTS

V.L. is supported by a FRIA (Fonds pour la Formation à la Recherche dans l'Industrie et dans l'Agriculture) fellowship of the F.R.S.-FNRS. F.K. is a research associate of the F.R.S.-FNRS. Computational resources were provided through two grants to D.B. (University of Liège "Crédit de démarrage 2012" SFRD-12/04; F.R.S.-FNRS "Crédit de recherche 2014" CDR J.0080.15) and by the Consortium des Équipements de Calcul Intensif (CÉCI) funded by the F.R.S.-FNRS.

V.L. conceived the study and designed experiments, performed experiments, analyzed the data, drafted and drew the figures, wrote the manuscript and approved the final manuscript. D.B. conceived the study and designed experiments, analyzed the data, wrote and reviewed the manuscript and approved the final manuscript. F.K. conceived the study, analyzed the data, reviewed the manuscript and approved the final manuscript. C.R., E.R., L.O., O.J., and C.M. performed experiments and approved the final manuscript.

## AUTHOR AFFILIATIONS

[1]InBioS-PhytoSYSTEMS, Eukaryotic Phylogenomics, University of Liège, Liège, Belgium
[2]InBioS, Center for Protein Engineering, University of Liège, Liège, Belgium

## AUTHOR ORCIDs

Valérian Lupo http://orcid.org/0000-0002-5532-2483
Denis Baurain http://orcid.org/0000-0003-2388-6185

## FUNDING

| Funder | Grant(s) | Author(s) |
|---|---|---|
| FNRS \| Fonds pour la Formation à la Recherche dans l'Industrie et dans l'Agriculture (FRIA) | | Valérian Lupo |
| Fonds De La Recherche Scientifique - FNRS (FNRS) | | Frédéric Kerff |
| Fonds De La Recherche Scientifique - FNRS (FNRS) | CDR J.0080.15 | Denis Baurain |
| Universite of Liege | SFRD-12/04 | Denis Baurain |

## AUTHOR CONTRIBUTIONS

Valérian Lupo, Conceptualization, Data curation, Formal analysis, Funding acquisition, Investigation, Methodology, Project administration, Resources, Software, Supervision, Validation, Visualization, Writing – original draft, Writing – review and editing | Célyne Roomans, Conceptualization, Formal analysis, Funding acquisition, Methodology, Project administration, Resources, Supervision, Validation, Visualization, Writing – original draft, Writing – review and editing | Edmée Royen, Conceptualization, Formal analysis, Investigation, Supervision, Writing – original draft, Writing – review and editing | Loïc Ongena, Conceptualization, Formal analysis, Funding acquisition, Methodology, Project administration, Resources, Supervision, Validation, Visualization, Writing – original draft, Writing – review and editing | Olivier Jacquemin, Formal analysis | Coralie Mullender, Formal analysis | Frédéric Kerff, Conceptualization, Formal analysis, Investigation, Supervision, Writing – original draft, Writing – review and editing | Denis Baurain, Conceptualization, Formal analysis, Funding acquisition, Methodology, Project administration, Resources, Supervision, Validation, Visualization, Writing – original draft, Writing – review and editing

## DATA AVAILABILITY

Publicly available data sets, including all detailed YAML configuration files used with Forty-Two (111, 112) and classify-ali.pl (D. Baurain; https://metacpan.org/dist/Bio-MUST-Core), and a detailed command line log file can be found at https://doi.org/10.6084/m9.figshare.21641612.

## ADDITIONAL FILES

The following material is available online.

### Supplemental Material

**Supplemental Figures (mSystems01401-24-S0001.pdf).** Figures S1 to S36.
**Supplemental data (mSystems01401-24-S0002.pdf).** Description of the unsuccessful regulon analysis.
**Captions (mSystems01401-24-S0003.docx).** Captions to Tables S1 to S4.
**Table S1 (mSystems01401-24-S0004.xlsx).** Functional prediction of InterProScan along with the prediction of signal peptide type and the number of predicted transmembrane segments.
**Table S2 (mSystems01401-24-S0005.xlsx).** Detailed distribution patterns of retained OGs using the bacterial database or prokaryotic database.
**Table S3 (mSystems01401-24-S0006.xlsx).** Jackknife support values computed from the 1,000 replicates of species resampling under three phylogenetic models.
**Table S4 (mSystems01401-24-S0007.xlsx).** Different combinations of intergenic regions.

Open Peer Review

**PEER REVIEW HISTORY (review-history.pdf).** An accounting of the reviewer comments and feedback.

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
