## [Reviewer comments · mSystems]

Identification and Characterization of Archaeal Pseudomurein Biosynthesis Genes through Pangenomics

Valérien Lupo, Célyne Roomans, Edmée Royen, Loïc Ongena, Olivier Jacquemin, Coralie Mullender, Frederic Kerff, and Denis Baurain

Corresponding Author(s): Denis Baurain, Universite de Liege

Review Timeline:

Submission Date:	November 6, 2024
Editorial Decision:	December 9, 2024
Revision Received:	January 3, 2025
Accepted:	January 14, 2025

Editor: Oleg Igoshin

Reviewer(s): Disclosure of reviewer identity is with reference to reviewer comments included in decision letter(s). The following individuals involved in review of your submission have agreed to reveal their identity: damien p devos (Reviewer #1)

Transaction Report:

DOI: <https://doi.org/10.1128/msystems.01401-24>

Re: mSystems01401-24 (Identification and Characterization of Archaeal Pseudomurein Biosynthesis Genes through Pangenomics)

Dear Prof. Denis Baurain:

Revision Guidelines

Sincerely,
Oleg Igoshin
Editor
mSystems

Reviewer #1 (Comments for the Author):

Authors have satisfyingly addressed most of my comments. Minor comments:

P20. The statement " It is clear that both polymers were different in their early evolutionary state". Clarify with reference the sentence or remove. Plural states?

P18. Taxonomic distribution of MurT and GatD among bacteria suggests that *C. exile* can be monoderm because it might belong

to Terrabacteria. Please notice that not all Terrab are monoderm.

P16-17. May be this is my misunderstanding, but the two statements appear to contradict each other: "Murdelta branches inside the MurE clan, within bacteria." followed, 5 lines later by "archaeal sequences (Murs including delta) never branch within bacterial sequences. Please clarify.

P15, please clarify that you take a 2D-ToL with both archaea and bacteria originating separately from the LUCA.

Reviewer #2 (Comments for the Author):

This is a revision of a manuscript describing an unbiased approach to identifying candidate ORFs in Methanopyrales and Methanobacteriales involved in PM biosynthesis with a specific focus on their origin and relationship to extant PG biosynthesis genes in bacteria.

My original assessment was that the manuscript corroborated prior work that relied upon protein homology based approaches. Importantly, the approach taken by Lupo et al. contributes a more nuanced evolutionary model for the emergence of PG and PM biosynthesis genes. Lupo et al. also propose several experimentally testable hypotheses, which represents a meaningful contribution to this understudied research area.

The revised manuscript presents mostly the same data but is substantially restructured especially in the Discussion section and is much improved with regard to placing the findings in context with other recent publications. The discussion is also reduced to several salient points that much more clearly convey the conceptual advance to a wide audience.

As with my first review, I had no doubts that the data presented justify the conclusions drawn. I have the following suggestions that might improve the clarity of the author's ideas, especially to a broader archaeal biology readership who would benefit from the author's findings. These are suggestions only and I would not need to review a second revision.

Line 137: "support" would be more appropriate than "confirm"

Line 148: What is the rationale for restricting the study to ten archaeal genomes?

Line 156: It is confusing to me why the 56 OGs not found in all five PM-containing archaea are quantified here if they are only to be excluded for the same criteria in line 219. It might be helpful to move Fig S1 to the main text and include annotation of inclusion/exclusion criteria at each stage to this figure so that the reader can better follow the bioinformatic workflow, because from the text alone I found this very difficult to follow.

Line 168: It is unclear what is meant by "annotated proteins belong to different pathways". Do you mean that these ORFs belong to HMM families that strongly support a biological function other than PM biosynthesis? The authors state their conclusions about these candidate gene clusters in this section but not the bioinformatic evidence underlying them. Is the reader meant to assume that these OGs should be excluded from consideration as candidate PM-associated loci, and why?

Line 203: It is unclear here why the prediction of signal peptides or TM helices in these proteins is important both in general and to the interpretation of the genetic context of these ORFs.

Line 237: For the sake of clarity, the hypotheses being tested in this section particularly should be explicitly introduced either here or in the introduction. In the introduction (lines 62-77 and 114-128) the two competing models for the evolution of PM are only alluded to, so a few sentences might go a long way to clarify the rationale for this approach.

Reviewer 1

Authors have satisfyingly addressed most of my comments. Minor comments:
P20. The statement " It is clear that both polymers were different in their early evolutionary state". Clarify with reference the sentence or remove. Plural states?

>>> Done. Thank you.

P18. Taxonomic distribution of MurT and GatD among bacteria suggests that *C. exile* can be monoderm because it might belong to Terrabacteria. Please notice that not all Terrab are monoderm.

>>> Indeed, you are correct. We have deleted this sentence because it was too sweeping.

P16-17. May be this is my misunderstanding, but the two statements appear to contradict each other: "Murdelta branches inside the MurE clan, within bacteria." followed, 5 lines later by "archaeal sequences (Murs including delta) never branch within bacterial sequences. Please clarify.

>>> Again, you are correct. We have added two adverbs ("exceptionally" and "almost") to alleviate the apparent contradiction.

P15, please clarify that you take a 2D-ToL with both archaea and bacteria originating separately from the LUCA.

>>> Done. We have added those sentences in the introduction: *CPS is a well-studied enzyme that has been used to root the "three-domain" tree of life because CarB results from an internal gene duplication that occurred before the Last Universal Common Ancestor (LUCA) (44-46). Nowadays, the most commonly held view is a "two-domain" tree of life with monophyletic Bacteria and paraphyletic Archaea from which emerge Eukarya (see 47), even though other models, such as a "one-domain" tree of life in which LUCA was a bacterium (48), are still possible.*

In addition, we have added the phrase "*two-domain tree of life*" at lines 513 and 537 of the Discussion section to underline our assumption.

Reviewer 2

This is a revision of a manuscript describing an unbiased approach to identifying candidate ORFs in Methanopyrales and Methanobacteriales involved in PM

biosynthesis with a specific focus on their origin and relationship to extant PG biosynthesis genes in bacteria.

My original assessment was that the manuscript corroborated prior work that relied upon protein homology based approaches. Importantly, the approach taken by Lupo et al. contributes a more nuanced evolutionary model for the emergence of PG and PM biosynthesis genes. Lupo et al. also propose several experimentally testable hypotheses, which represents a meaningful contribution to this understudied research area.

The revised manuscript presents mostly the same data but is substantially restructured especially in the Discussion section and is much improved with regard to placing the findings in context with other recent publications. The discussion is also reduced to several salient points that much more clearly convey the conceptual advance to a wide audience.

>> Thank you for the positive appreciation of our work.

As with my first review, I had no doubts that the data presented justify the conclusions drawn. I have the following suggestions that might improve the clarity of the author's ideas, especially to a broader archaeal biology readership who would benefit from the author's findings. These are suggestions only and I would not need to review a second revision.

Line 137: "support" would be more appropriate than "confirm"

>>> Done.

Line 148: What is the rationale for restricting the study to ten archaeal genomes?

>>> We have chosen organisms that have been empirically shown to have pseudomurein. This is now precised in the Material and Methods section (with the relevant citations from the Bergey's Manual). In addition, starting or pangenomics from a small number of organisms allowed us to manually address potential errors, such as wrong gene prediction (see also our answer to your comment just below).

Line 156: It is confusing to me why the 56 OGs not found in all five PM-containing archaea are quantified here if they are only to be excluded for the same criteria in line 219. It might be helpful to move Fig S1 to the main text and include annotation of inclusion/exclusion criteria at each stage to this figure so that the reader can better follow the bioinformatic workflow, because from the text alone I found this very difficult to follow.

>>> This is in line with the previous comment. In some genomes, gene prediction errors may occur, and with a “strict” pangenomic approach (i.e., considering only genes present in all selected genomes) could result in missing such genes. To address this issue, we opted for a more “loose” pangenomic approach. This method involves initially collecting potential orthologous groups (OGs) using different taxonomic filters, confirming (or infirming) the absence of (mispredicted) genes, and ultimately discarding those OGs that do not meet the criteria (as illustrated in Fig. S1).

This approach is illustrated in the section “Taxonomic distribution of candidate proteins and their homologues” of the Results, in which we have stated the following: *As an illustration, in Methanothermobacter thermoautotrophicus str. Delta, OG0001472 and Murδ had been annotated as pseudogenes. However, in PM-containing archaea, the synteny of these genes from cluster A is highly conserved. Thus, our “loose” pangenomic approach rightfully led us to rescue OGs that would have been missed by a stricter approach. After trying to complete all the OGs, we retained only those containing protein sequences from all five PM-containing archaea, decreasing their number from 111 to 49.*

As bioinformaticians we certainly appreciate detailing our methodology. However, considering the broader readership of our manuscript, we deemed it more appropriate to leave Fig. S1 in the Supplementary Materials.

Line 168: It is unclear what is meant by “annotated proteins belong to different pathways”. Do you mean that these ORFs belong to HMM families that strongly support a biological function other than PM biosynthesis? The authors state their conclusions about these candidate gene clusters in this section but not the bioinformatic evidence underlying them. Is the reader meant to assume that these OGs should be excluded from consideration as candidate PM-associated loci, and why?

>>> We have modified the sentence for the sake of clarity, but the bioinformatic evidence was already cited in Table S1.

Line 203: It is unclear here why the prediction of signal peptides or TM helices in these proteins is important both in general and to the interpretation of the genetic context of these ORFs.

>>> Granted, the exact justification for these analyses was missing. We have restructured this paragraph and added the following sentence at the end to improve its clarity: *Overall, these targeting analyses are useful to gain insight into the potential position of each protein in the PM biosynthetic pathway.*

Line 237: For the sake of clarity, the hypotheses being tested in this section particularly should be explicitly introduced either here or in the introduction. In the introduction

(lines 62-77 and 114-128) the two competing models for the evolution of PM are only alluded to, so a few sentences might go a long way to clarify the rationale for this approach.

>>> You are totally correct. To address this comment, we have added the following sentences at the end of the Introduction: *We also investigated the evolutionary origins of shared PM and PG biosynthesis genes, with the aim to distinguish between two main hypotheses for the emergence of PM in class I methanogens: vertical inheritance from LUCA followed by losses in most archaeal lineages or convergent evolution through HGT from Bacteria. To this end, we performed phylogenetic analyses of the Mur domain-containing family, the ATP-grasp superfamily and the MraY-like family, using multiple variations of the taxon sampling and different AA substitution models.*

Re: mSystems01401-24R1 (Identification and Characterization of Archaeal Pseudomurein Biosynthesis Genes through Pangenomics)

Dear Prof. Denis Baurain:

Your manuscript has been accepted, and I am forwarding it to the ASM production staff for publication. Your paper will first be checked to make sure all elements meet the technical requirements. ASM staff will contact you if anything needs to be revised before copyediting and production can begin. Otherwise, you will be notified when your proofs are ready to be viewed.

Sincerely,
Oleg Igoshin
Editor
mSystems